# Treatment Response Biomarkers in Asthma and COPD

**DOI:** 10.3390/diagnostics11091668

**Published:** 2021-09-13

**Authors:** Howraman Meteran, Pradeesh Sivapalan, Jens-Ulrik Stæhr Jensen

**Affiliations:** 1Department of Internal Medicine, Respiratory Medicine Section, Copenhagen University Hospital—Herlev and Gentofte, 2900 Hellerup, Denmark; pradeesh.sivapalan.02@regionh.dk (P.S.); jens.ulrik.jensen@regionh.dk (J.-U.S.J.); 2Department of Microbiology and Immunology, University of Copenhagen, 1353 Copenhagen, Denmark; 3Department of Internal Medicine, Zealand University Hospital, 4000 Roskilde, Denmark; 4Department of Clinical Medicine, Faculty of Health Sciences, University of Copenhagen, 1353 Copenhagen, Denmark

**Keywords:** COPD, asthma, biomarker treatment, response

## Abstract

Chronic obstructive pulmonary disease (COPD) and asthma are two of the most common chronic diseases worldwide. Both diseases are heterogenous and complex, and despite their similarities, they differ in terms of pathophysiological and immunological mechanisms. Mounting evidence supports the presence of several phenotypes with various responses to treatment. A systematic and thorough assessment concerning the diagnosis of both asthma and COPD is crucial to the clinical management of the disease. The identification of different biomarkers can facilitate targeted treatment and monitoring. Thanks to the presence of numerous immunological studies, our understanding of asthma phenotypes and mechanisms of disease has increased markedly in the last decade, and several treatments with monoclonal antibodies are available. There are compelling data that link eosinophilia with an increased risk of COPD exacerbations but a greater treatment response and lower all-cause mortality. Eosinophilia can be considered as a treatable trait, and the initiation of inhaled corticosteroid in COPD patients with eosinophilia is supported in many studies. In spite of advances in our understanding of both asthma and COPD in terms pathophysiology, disease mechanisms, biomarkers, and response to treatment, many uncertainties in the management of obstructive airways exist.

## 1. Introduction

Chronic obstructive pulmonary disease (COPD) and asthma affect more than 600 million individuals globally. Although COPD and asthma are heterogenous and complex diseases that share similarities concerning symptoms, inflammation, and airflow limitation, they also differ with regard to certain key features. Asthma is characterized by variable airflow limitation, airway hyperresponsiveness, and airway inflammation [1]. It has several phenotypes with distinct aetiologies that can be classified based on triggers, clinical presentation, and inflammatory type [2]. Asthma was previously thought to be characterized by eosinophilic inflammation, but it is now recognized that characteristics of asthma can be present without eosinophilic inflammation [2,3]. COPD is characterized by chronic inflammation of the lungs with the presence of neutrophils, macrophages, and T-lymphocytes. The T-lymphocytes consist mostly of TH1, TH17, and cytotoxic T cells [4]. Based on the idea that COPD is characterized by neutrophilic inflammation, the presence of eosinophilic airway inflammation has been used to discriminate between asthma and COPD. However, this notion is flawed because around 40% of COPD patients exhibit eosinophilic inflammation even when adjusting for asthma [5]. These patients exhibit characteristics similar to those of asthma patients, and the term asthma–COPD overlap (ACO) has been used to describe this subset of patients who simultaneously exhibit characteristics of both diseases [6].

This overlap between asthma and COPD presents many challenges, as different aetiologies with different treatment needs can present with similar symptoms. Thus, distinguishing between allergic inflammation, bacterial infection, and other disease phenotypes is crucial to providing optimal treatment as well as reducing the burden and side effects of ineffective treatment.

The identification of the various phenotypes makes it possible to take an individualized approach to clinical decision making when treating patients with asthma and COPD. The need for predictive biomarkers has become increasingly vital to identifying patients who are most likely to achieve clinical benefit and minimum side effects [7]. Thus, the aim of this study is to review the identified treatment response biomarkers in asthma and COPD.

## 2. Biomarkers in Asthma

Asthma has long been classified as either atopic (“extrinsic”) or non-atopic (“intrinsic”) asthma but is now considered an umbrella diagnosis for various diseases with different endotypes (underlying immunological mechanisms) and phenotypes (e.g., atopy, obesity and age). Cluster analyses of data from large cohorts have led to two major asthma endotypes, T2-high and non-T2-high asthma [8,9].

### 2.1. T2-High Asthma

Reduced barrier function in the respiratory epithelium seems to play an important role in T2-high asthma, as microbes and allergens activate epithelial-derived alarmins, such as thymic stromal lymphopoietin (TSLP), interleukin (IL)-25, and IL-33 [10]. Activation of these upstream cytokines results in several type 2 immune responses. TLSP activates T- and B-cell responses, while IL-25 and IL-33 activate innate lymphoid cells (ILCs), which play a crucial role in the production of IL-5 and IL-13 [11]. The principal cytokines produced by Th2-cells are IL-4, IL-5, and IL-13 [12]. These cytokines are involved in numerous immunological processes, such as the production of downstream cytokines (IL13 and IL-4), activation of mast cells (IL-13), B-cell activation to undergo immunoglobulin E (IgE) isotype switching (IL-4), and maturation and survival of eosinophils (IL-5) [13] and subsequently promote pathophysiological changes, such as increased mucus secretion, airway hyperresponsiveness, inflammation, and tissue remodelling [3,14]. Thus, both Th2-cells and ILCs are involved in type 2 inflammation, which is manifested as high IgE and eosinophils in asthma.

Corticosteroids are the cornerstone of asthma treatment and result in clinically significant improvements in lung function, symptom control, and reduction of asthma exacerbations [15]. However, approximately 10% of asthma patients remain uncontrolled in terms of symptoms and exacerbations despite treatment with high-dose, inhaled corticosteroids in combination with long-acting β-2-agonists and long-acting muscarinic-antagonists. For this group of patients, treatment with monoclonal antibodies targeting some of the cytokines can significantly improve various clinical outcomes in patients with severe asthma [16]. Omalizumab (humanized, monoclonal antibody that binds to circulating IgE molecules) was the first approved treatment for patients with allergic asthma, followed by treatments targeting the IL-5 pathway: mepolizumab (monoclonal antibody that binds to IL-5), reslizumab (also a monoclonal antibody that binds specifically to IL-5), and benralizumab (monoclonal antibody that binds directly to the IL-5-receptor*α* on eosinophils) [17]. Dupilumab is the most recently approved biological treatment for severe asthma. Dupilumab blocks the α-subunit of the IL-4-receptor, which is used by both IL-4 and IL-13 and thus inhibits signal transduction from these key mediators of type 2 inflammation [18].

#### 2.1.1. Eosinophilia

The prevalence of eosinophilic inflammation among the asthmatic population is 50% but might be underestimated [19]. Eosinophils are the most important cells associated with type 2 inflammation, and when activated, they release a number of inflammatory mediators from intracellular granules [20], resulting in airway remodelling and bronchoconstriction [21]. Eosinophil inflammation in asthma is associated with poor prognosis [22] and, moreover, predicts response to treatment with corticosteroids [23]. Thus, several biomarkers have been identified and utilized to quantify eosinophilic airway inflammation and will be discussed in the following sections.

##### Sputum and Blood Eosinophils

Eosinophilic airway inflammation in induced sputum (cut-off > 3%) is considered to be a more accurate biomarker of T2-inflammation than absolute eosinophil count in peripheral blood [24]. An inconsistent association between airway and peripheral eosinophils has been observed in several studies and might be due to heterogenous study populations [25,26]. The correlation between sputum and blood eosinophils was investigated in a clinical study including both asthma and COPD patients and showed that the correlation was better in the asthmatic population [27]. Another study using data from the SPIROMICS (Subpopulation and Intermediate Outcome Measures In COPD Study) cohort found that stratification by sputum eosinophils but not blood eosinophils was associated with an increased risk of COPD exacerbations [28]. However, the presence of blood eosinophils in both asthma and COPD is associated with accelerated lung function decline and increased risk of exacerbations [29,30,31] and serves as a useful biomarker to identity patients with severe eosinophilic asthma [32] and response to inhaled corticosteroids in COPD [33].

Moreover, as induced sputum for daily clinical practice is laborious and can be bothersome for the patient, the use of blood eosinophils as a marker of T2-inflammation is more widely used.

Various cut-off values between 150–400 cells/µL have been used in the definition of blood eosinophilia and are able to predict response to treatment with anti-IL-5 in asthmatic individuals [34,35,36]. However, mounting evidence suggests that eosinophil count should be deemed a continuous variable and that higher levels predict a greater response [33].

Furthermore, a post-hoc study examining the stability of blood eosinophils in asthmatic individuals found that a single measurement might be insufficient in the diagnosis and management of asthma and that the instability was more pronounced for eosinophil counts between 150–299 cells/µL [37]. 

A study comprising stable COPD patients showed that using a threshold of ≥300 cells/µL in peripheral blood enabled the identification of sputum eosinophilia in 71% of the patients [38].

##### Eosinophils and Response to Treatment

Based on the last two decades of research, it is now established that eosinophilia can be used as a predictive biomarker for both the initiation and discontinuation of treatment with inhaled corticosteroids [39,40,41,42]. A retrospective study of asthmatics from a secondary care centre showed that inhaled corticosteroids (ICS) reduced both sputum and blood eosinophils, and clinical improvements were observed in terms of quality of life, forced expiratory volume in first second (FEV1), airway hyperresponsiveness, and exacerbation rate in those asthma patients with eosinophilic inflammation [42]. In addition, reducing ICS among patients with non-eosinophilic inflammation resulted in improved asthma control. In a 16-week trial with mild-to-moderate asthmatic individuals, sputum eosinophil counts two weeks after discontinuation of ICS and the change in eosinophil counts from before and after cessation of ICS predicted subsequent loss of asthma control [39]. A Cochrane review concluded that the frequency of asthma exacerbations can be reduced by tailoring the asthma treatment based on sputum eosinophils [41].

Randomized clinical trials have shown that a higher baseline eosinophil count predicts a greater reduction of severe asthma exacerbations in patients treated with inhaled corticosteroids [43]. A systematic review and meta-analysis including 61 studies found that oral corticosteroids improved lung function and reduced asthma symptoms and markers of type 2 inflammation and that patients with increased sputum and blood eosinophils at baseline were more responsive to treatment with oral corticosteroids [44]. In a clinical study of OCS-dependent asthmatic individuals, several type-2-related biomarkers returned to baseline levels after month after treatment (0.5 mg/kg prednisolone for 7 days) [45].

The level of blood eosinophils is also the key biomarker in selecting patients for treatment with monoclonal antibodies targeting the IL-5 pathway. In both the DREAM (Mepolizumab For Severe Eosinophilic Asthma) and MENSA (Mepolizumab Treatment for Patients With Severe Eosinophilic Asthma) studies, an eosinophil count of ≥150 cells/µL at baseline predicted a better response to treatment with mepolizumab in severe asthma patients [35]. The pooled analyses of the CALIMA (randomized, double-blind, placebo-controlled phase 3 trial with benralizumab) and SIROCCO (randomized, multicentre, placebo-controlled phase 3 trial with benralizumab) studies showed that in severe asthma patients, blood eosinophils ≥300 cells/µL and exacerbations in the previous year were predictors of a greater treatment response compared with those patients with eosinophils <300 cells/µL [46].

The early efficacy studies on reslizumab included patients with persistent asthma and blood eosinophils ≥400 cells/µL and observed significant improvements in the annual frequency of asthma exacerbations [47] and FEV1 [48].

#### 2.1.2. Fraction of Exhaled Nitrogen Oxide (FeNO)

Nitric oxide is produced in the bronchial airway by inducible nitric oxide synthase (iNOS) and is mediated by type 2 inflammatory cytokines, such as IL-4 and IL-13 [49]. The measurement of FeNO is widely used as a marker of eosinophilic inflammation in the airways [50], and when combined with blood eosinophils, it might be useful for differentiating between COPD and asthma–COPD overlap [51].

A study including steroid-naive asthma patients and healthy individuals found that the level of FeNO was highest among patients with allergic asthma, followed by non-allergic asthma and healthy individuals [52]. FeNO is associated with reversibility of airway obstruction, blood eosinophils [52], and bronchial hyperresponsiveness [53]. Various cut-off values have been suggested, but it has become evident that values below 25 ppb are not associated with eosinophil inflammation, whereas values above 50 are strongly associated with eosinophilic inflammation (Table 1) [54].

A number of factors, such as sex, smoking, atopy, BMI, and chronic rhinosinusitis with nasal polyps, even in the absence of asthma, can affect the level of FeNO and should be taken into account in clinical evaluation [55,56].

A three-year follow-up study including severe asthma patients found an association between patients with sustained high levels of FeNO (≥50 ppb) and an increased risk of asthma exacerbations (shorter exacerbation-free survival time and number of exacerbations) compared with those with sustained low levels of FeNO (<25 ppb) [57]. Another study that used data from the same cohort found that among a number of type-2-related biomarkers, only FeNO was associated with an increased risk of exacerbations [58]. Interestingly, the association was independent of past exacerbation status.

An early randomized, placebo-controlled trial showed that treatment with inhaled corticosteroids significantly decreased the level of FeNO compared with placebo after only two weeks of treatment [59]. The decreased level was sustained during the treatment period (four weeks) and increased significantly after a (two-week) washout period. Another randomized, open-label clinical trial examined FeNO in relation to spirometry and sequential changes in relation to inhaled corticosteroids [60] and found that FeNO but not spirometry was able to differentiate between patients treated with or without ICS. Moreover, the reduction in FeNO correlated significantly with patients’ adherence to ICS.

In a more recent double-blind, randomized, placebo-controlled multicentre study, the baseline level of FeNO predicted the response to treatment with extrafine ICS in terms of significant changes in Asthma Control Questionnaire 7 (ACQ-7): for every 10-ppb increase in baseline FeNO, the change in ACQ-7 was greater in the treatment group than in the placebo group (difference between groups 0.071 (0.002–0.139), *p* = 0.04) [61].

A number of randomized, clinical trials (RCTs) have assessed FeNO as a tool for guiding asthma treatment [62,63]. In an Australian RCT including 220 pregnant, non-smoking asthmatic women, the exacerbation rate was lower in the FeNO-group than in the control group in which the treatment was adjusted according to clinical symptoms, incidence rate ratio 0.47 (0.33–0.76), *p* = 0.001 [62]. A 36-week RCT including 80 asthmatic Danish adults assessed the utility of a FeNO-guided versus symptom-based treatment algorithm [63]. The decrease in airway hyperresponsiveness (AHR) from week 8 to 24 was significantly different in the FeNO-group compared with the control group, suggesting that the use of FeNO resulted in an earlier lowering of AHR. However, no differences were observed in week 36.

The data to support a universal use of FeNO to tailor the asthma treatment are lacking, but FeNO might be useful for guiding treatment for asthma patients with frequent exacerbations. In the most recent Cochrane review on this subject, Petsky et al. found significant differences in asthma exacerbations between the FeNO-group versus non-FeNO-group, rate ratio 0.59 (0.45–0.77), but no differences were observed for symptoms, lung function, or inhaled corticosteroids [64]. 

#### 2.1.3. Immunoglobin E (IgE)

It has been known for decades that the level of immunoglobin E (IgE) is elevated in allergic asthma patients compared with non-allergic asthma patients [65] and is significantly related to atopic status [66]. Longitudinal studies have shown an association between high levels of IgE and impaired lung function in both asthmatic individuals [67] and non-asthmatic and non-COPD individuals [68]. In a small study including nonsteroid-dependent asthmatic individuals, treatment with corticosteroids during an exacerbation resulted in an initial elevation of total IgE and a subsequent decrease [69], whereas serum IgG was decreased. In a more recent, 44-week randomized controlled trial, asthmatic individuals with baseline serum IgE ≥ 350 K/μL obtained greater benefit from inhaled corticosteroids compared with those with baseline serum IgE < 350 K/μL (Table 1) [70]. 

Treatment with anti-IgE (omalizumab) in patients with allergic asthma has been shown to significantly improve asthma control and lung function as well as reduce ICS use and exacerbations (Table 1) [71,72]. A high baseline level of IgE is a strong predictor of response to treatment with anti-IgE in both allergic asthma [73] and chronic spontaneous urticaria [74]. However, a small, retrospective case-control study showed that treatment response to omalizumab was similar in asthma patients with baseline IgE levels between 30 and 700 UI/mL and IgE levels >700 IU/mL [75].

Although immunological changes in terms of an initial increase in total IgE and decrease in free serum IgE had already been observed in the first clinical studies with omalizumab [76,77], the underlying mechanisms have yet to be fully elucidated [78,79]. However, it has been suggested that total IgE, after the initial accumulation, can serve as a biomarker to monitor IgE production and guide treatment on an individual level [80]. 

Similar changes in total IgE levels are observed upon treatment with allergen immunotherapy, but these changes are not related to the subsequent reduced response to an allergen [72], whereas an increase in immunoglobin G (especially IgG1 and IgG4) seems to play an important role in clinical outcomes, as these subclasses of antibodies compete with IgE in binding the specific allergen [81].

#### 2.1.4. Periostin

Periostin is a matricellular protein present in the extracellular matrix [82]. Periostin is considered to be a type-2-related biomarker [83] that is influenced by IL-4 and IL-13. Moreover, periostin differs from other type 2 inflammatory markers in that it is involved in airway remodelling and thus can be considered a chronic rather than an acute biomarker. Serum periostin has been shown to be a biomarker of persistent eosinophilic inflammation and fixed airflow limitation in asthmatic individuals treated with inhaled corticosteroids [84].

A randomized controlled trial comparing the effect of ICS on serum periostin level and the association with inflammation found that ICS significantly lowered serum periostin [85] and that the decrease in periostin was associated with improved lung function, decreased sputum eosinophils, and airway remodelling. The association between periostin and lung function has been confirmed in other clinical studies [86].

Anti-IL-13 has not yet been approved for the treatment of severe asthma. A phase 2, double-blind, placebo-controlled trial including 219 asthmatic adults found that lebrikizumab significantly increased FEV1 compared with placebo and that the improvement in lung function occurred only in patients with high baseline serum periostin [87]. These findings are in line with subgroup analyses from other large RCTs with anti-IL-13. In a 52-week RCT with tralokinumab, patients with pre-treatment levels of periostin had improvements in asthma exacerbation rate, lung function, and ACQ-6 [88].

### 2.2. Non-T2-High Asthma

Although many asthmatic patients have signs of type 2 inflammation, a large group of patients does not. Roughly half of asthma patients show no signs eosinophilic inflammation [15], and this inflammatory state seems to be stable for at least five years [89]. Non-eosinophilic asthma is recognized as a diverse phenotype. It is described as neutrophilic asthma if neutrophils are elevated in sputum and as pauci-granulocytic asthma if neither neutrophils nor eosinophils are elevated [15]. Non-eosinophilic asthma is also associated with a poor response to inhaled and oral corticosteroid treatment [89,90]. Furthermore, it was shown that the ICS dose could be reduced in two-thirds of non-eosinophilic patients (defined as sputum eosinophils <3% and blood eosinophils <400/μL) [91].

Neutrophilic inflammation is a phenotype where the TH2-driven response is replaced by a TH17-driven inflammatory response. This phenotype is only recently being acknowledged and was previously thought to be a misdiagnosis of COPD or induced by corticosteroid treatment because it promotes neutrophilic inflammation [92]. No clear definition of neutrophilic asthma exists, and studies in healthy individuals have shown that the normal range of neutrophils in induced sputum is between 30–50% [93]. Multiple measurements and a cut-off value above 5 × 10^9^/L are suggested [94]. Furthermore, as airway neutrophilia is related to age in adults, the use of age-specific reference values is recommended [95]. Neutrophilia is present in 20–30% of the asthmatic population, although the prevalence varies across regions [96]. This phenotype is more often associated with smoking [97], obesity [98], and various forms of pollution, air pollution, ozone, and other pollutants [99,100]. Neutrophilic inflammation may also be caused by acute airway infections [101], particularly in children [102]. Elevated neutrophil counts have also been associated with a decrease in microbial diversity [103].

A range of treatment strategies has been suggested for this subset of patients, with varying efficacy [91]. Smoking cessation has been shown to benefit asthma patients independent of inflammatory phenotype, but it could be hypothesized that this benefit would be even more significant for patients with neutrophilic inflammation [104]. Obesity has also been associated with neutrophilic asthma, and weight loss has been shown to reduce asthmatic symptoms without changes in inflammatory markers, indicating an inflammation-independent mechanism for asthma [98,105]. TH2-blocking drugs, such as anti-IL-5 or anti-IgE, are not viable for non-eosinophilic asthma. A range of other specific biologics-targeting cytokines, such as TNFα, IL-1, IL-6, and IL-17, have been tried, but none have yet seen widespread use, and more research is needed to identify safe and effective drugs [106]. Systemic inflammation has been associated with neutrophilic asthma, and a microarray analysis has revealed more than 400 genes that are involved in IL-1, TNF-α/nuclear factor-κB, and Kit receptor pathway [107]. These findings may enable a novel treatment strategy for neutrophilic asthma. The AMAZES study (Effect of Azithromycin on Asthma Exacerbations and Quality of Life) showed that oral azithromycin significantly decreased exacerbations and improved quality of life in patients with uncontrolled asthma despite medium-to-high-dose inhaled corticosteroids + long acting β-2 agonist (LABA) [108]. A recent analysis in an AMAZES sub-population showed that baseline sputum TNF-receptors 1 and 2 were significantly increased in neutrophilic versus non-neutrophilic asthma and were related to increased age, lower lung function, and worse asthma control [109]. Azithromycin significantly reduced sputum TNF-receptor 2 and TNF compared with placebo, particularly in non-eosinophilic asthma.

## 3. Biomarkers in COPD

### 3.1. Eosinophil Counts to Guide Use of Systemic Corticosteroids

As seen in asthma, increased levels of blood eosinophils have been associated with worse clinical outcomes in a range of parameters, such as length of hospital admission, risk of readmission, and risk of future exacerbation [110,111,112,113]. Furthermore, blood eosinophil counts predict response to corticosteroids [31,114]. Systemic corticosteroids, such as prednisolone, are used in the treatment of acute exacerbations. These drugs can alleviate symptoms in many cases, but they do not reduce mortality [115]. Corticosteroids are, however, associated with significant side effects [116]. For this reason, there has recently been a great deal of interest in using blood eosinophil counts to predict which patients will benefit from systemic corticosteroid treatment. One study categorized exacerbation as being either eosinophilic or non-eosinophilic and showed non-inferiority when replacing systemic corticosteroids with placebo in the non-eosinophilic group. This algorithm reduced total corticosteroid use by 49% in the eosinophil-guided group [117]. The CORTICO-COP trial (Eosinophil-guided Corticosteroid Therapy In Patients Admitted To Hospital With COPD Exacerbation) used daily blood eosinophil counts to guide corticosteroid use and compared these patients to a control group receiving a five-day prednisolone treatment. This algorithm led to a reduction of median treatment duration from five to two days with no change in the number of days alive and out of hospital within 14 days or the 30-day mortality [118]. The authors also reported a decreased risk of worsening of pre-existing diabetes in the eosinophil-guided group compared to the control group.

### 3.2. Eosinophil Counts to Guide Use of Inhaled Corticosteroids

While systemic corticosteroids are typically used to treat patients admitted with acute exacerbations, ICS are often used to treat COPD patients to alleviate symptoms and reduce risk of exacerbation [119]. While ICS are generally thought to be safer than systemic corticosteroids, they have been associated with an increased risk of infection and pneumonia [120]. ICS have also been indicated to cause many of the known systemic side effects of oral corticosteroids, such as cataracts [121,122] and bone demineralization. No randomized clinical trials have currently shown that blood eosinophil counts can be used to guide prescription of ICS. However, several post-hoc studies of large clinical trials have shown that blood eosinophil count might be used as a predictor of response to treatment with ICS [123,124] These results have made it into GOLD guidelines 2019, which recommend ICS only for patients with blood eosinophil counts ≥100 /μL [125]. A post-hoc analysis of three randomized controlled trials found a greater response to ICS/LABA compared with both placebo and long acting muscarine antagonist (LAMA) in COPD patients with baseline blood eosinophils ≥2% [126]. In another study, Siddiqui and colleagues conducted post-hoc predictive modelling using data from the FORWARD trial and externally validated data to examine the association between baseline eosinophils and the effect of ICS in exacerbations [127]. The authors found that ICS/LABA across all eosinophil levels reduced COPD exacerbations compared with LABA, and the difference in exacerbation rate was more pronounced with increasing level of baseline blood eosinophils. Two studies have investigated the safety of withdrawing ICS, and post-hoc analyses have been done on the effects of blood eosinophil counts. The SUNSET study found negative effects on lung function when withdrawing ICS from long-term ICS/LAMA/LABA treatment, but they found that the negative effects of withdrawal were much less apparent in patients with blood eosinophil counts of ≤300/μL [128]. The WISDOM trial showed no difference in the risk of COPD exacerbation when withdrawing ICS [129]. However, a post-hoc analysis of the WISDOM trial showed a higher exacerbation rate after ICS withdrawal in patients with blood eosinophil counts ≥300/μL [130]. There is currently one ongoing multi-centre trial, the COPERNICOS trial (NCT04481555), which aims to assess whether blood eosinophil counts can be used to guide ICS usage in patients with severe or very severe COPD, the results of which are expected by 2025 [131]. The intervention group will have their ICS adjusted every three months based on whether blood eosinophils count is ≤300/μL with the aim of reducing corticosteroid exposure while being non-inferior to the current treatment regime (COPERNICUS).

### 3.3. IL-5-Targeted Therapy and Blood Eosinophils

Since a subset of COPD patients present with eosinophilic airway inflammation, it can be hypothesized that these patients may benefit from the same therapies used in the treatment of asthma. In the METREX and METREO randomized trials, the efficacy and safety of mepolizumab were investigated in a population of COPD patients [132]. In METREX, patients were randomized to placebo or 100 mg mepolizumab every four weeks. In METREO, patients were randomized to either placebo, 100 mg or 400 mg every five weeks. Neither trial was able to show a significant difference in exacerbation rates or secondary outcomes in the whole population, but a post-hoc analysis on patients from both trials with blood eosinophils ≥300 /μL showed a statistically significant reduction of exacerbation rate by 23% in patients receiving 100 mg mepolizumab versus placebo. Future studies investigating the effect of mepolizumab in the subgroup of COPD patients are warranted.

A phase 2a randomized trial investigated whether benralizumab could reduce acute exacerbation in COPD with ≥ 3% eosinophils in sputum [133]. No statistically significant effect was seen on exacerbation rates, but the sample size of the study was low, and a numerical reduction was seen in patients with blood eosinophils ≥200/μL. The GALATHEA (Benralizumab Efficacy in Moderate to Very Severe COPD With Exacerbation History) and TERRANOVA (Benralizumab for the Prevention of COPD Exacerbations) randomized phase 3 trials further investigated the effect of benralizumab and were unable to show a reduction in exacerbation frequency in patients with moderate to very severe COPD with blood eosinophil counts ≥220/μL. To date, no trials with reslizumab in COPD patients have been conducted.

Thus, there are no current indications that IL-5-targeted therapies are a viable strategy for treating COPD.

## 4. Procalcitonin as a Tool to Guide Antibiotic Treatment for Respiratory Tract Infections

Procalcitonin (PCT) is a fast-acting biomarker of bacterial infection [134]. Thus, it has been suggested that PCT could be used in the early detection of bacterial infections and to guide antibiotic therapy [135]. Antibiotic resistance is a growing concern and threatens both human health and food security. While the causes of resistance are many, overuse of antibiotics in humans and livestock is a central component in accelerating the development of resistance [136]. It is therefore crucial to administer antibiotics only when the expected clinical benefits overweigh the risk of side effects and resistance. Bacteria can be isolated from sputum in 40–60% of acute exacerbations of COPD, while acute exacerbations of asthma are more commonly triggered by other factors, such as allergens, gastro-oesophageal reflux, and viral infections, though bacterial infection occasionally occurs as well [137].

Although bacterial infections are uncommon in asthma exacerbations, antibiotics are heavily prescribed. From 2006 to 2012, 51% of patients admitted with asthma to 383 US hospitals were prescribed antibiotics [138]. Two clinical trials have looked at PCT-guided therapy, specifically in the context of asthma exacerbations.

PCT has been used to guide antibiotic usage in both mild [139] and severe [140] asthma exacerbations without being inferior on clinical outcomes. Both studies used a protocol where antibiotics were strongly discouraged when PCT was below 0.1 μg/L, discouraged when between 0.1 μg/L and 0.25 μg/L, and encouraged above 0.25 μg/L. Antibiotics were prescribed to 80–90% of the control group but to only around half of the intervention group, leading to a significant reduction in antibiotics. Although bacterial infections in COPD are estimated to account for roughly half of acute exacerbations, antibiotics are commonly prescribed, along with systemic corticosteroids, to most patients presenting with moderate to severe acute exacerbations [31]. This suggests that antibiotics are overprescribed. Corticosteroids, on the other hand, are known to increase the risk of pneumonia [120], so it could be hypothesized that systemic corticosteroids are ineffective or even harmful in patients with acute exacerbations caused by bacterial airway infections. Research on this topic is limited, but one study found that corticosteroids did not improve outcomes in patients with diagnosed pneumonia [141]. This finding accords with the previously discussed fact that corticosteroid response is worse in patients presenting with neutrophilic inflammation, which is the typical inflammatory pattern of bacterial infections [142].

A patient-level meta-analysis on 26 trials including a total of 6708 patients investigated whether PCT was useful as a tool to guide antibiotic usage in all airway infections. This analysis showed that PCT-guided therapy was effective in reducing antibiotic prescription rates, antibiotic side effects, and even mortality (Table 1) [143].

A systematic review from 2017 of 32 RCTs examining procalcitonin to guide antibiotic usage in airway infections in general found that procalcitonin-guided algorithms led to lower mortality, shorter duration of antibiotic treatment, and fewer antibiotic side effects [144].

## 5. Conclusions

Asthma and COPD compose a great burden for public health worldwide, and the need for early identification of cases and intervention is still of great importance. Obstructive airway diseases are a group of complex and heterogeneous diseases. Our knowledge has evolved markedly due to research over the past 15 to 20 years, and several distinct phenotypes and endotypes are now described in the literature. The advantages of conducting a systematic assessment of asthma and COPD become evident when specific and reliable biomarkers enable the possibility of precision medicine. While type 2-high asthma is the most well-described endotype, our understanding of non-type-2 is now increasing, and if specific biomarkers for non-type 2 asthma can be identified, it is reasonable to expect better treatment options for this group of patients. The role of eosinophilia in COPD is now established as a valuable biomarker to predict both prognosis and treatment response and serves as a tool for tailoring the use of corticosteroids for maintenance therapy and during an exacerbation. To date, evidence for the clinical benefits of monoclonal antibodies in COPD-patients has not been overwhelming. However, future studies using biomarkers to identify patients who are most likely to respond are warranted.

## Figures and Tables

**Table 1 diagnostics-11-01668-t001:** Summary table of biomarkers in asthma and COPD.

Biomarker	Method of Measurement	Indicates	Predicts
**Eosinophilia**	Sputum and blood sampling	- Type 2 inflammation - Eosinophilic inflammation: - Various cut-off values have been suggested, but the vast majority suggest ≥3% in sputum or ≥300 cells cells/μL in blood. However, the general conception is that eosinophilia should be considered as a continuous rather than a binary biomarker	- Risk of exacerbations - Response to inhaled corticosteroids - Response to biologics (anti-IgE, anti-IL-5, and anti-IL-4R)
**Neutrophilia**	Sputum and blood sampling	- Neutrophilic inflammation - Non-type 2 inflammation - Associated with obesity and air pollution	- Smaller response to inhaled corticosteroids compared with eosinophilic inflammation
**FeNO**	- Non-invasive device to measure the concentration of fractional nitric oxide in exhaled breath	- Type 2 inflammation - Eosinophilic inflammation: - FeNO levels <25 ppb: normal (eosinophilia is unlikely) 25–50 ppb: intermediate (eosinophilia is possible) >50 ppb: high (eosinophilia is very likely)	- Exacerbation history - Response to inhaled corticosteroids - Response to anti-IgE and anti-IL-4R
**IgE**	Blood sample or skin prick test	- Atopy	- Response to inhaled corticosteroids (levels ≥350 K/μL) - Response to anti-IgE
**Periostin**	Blood sample	- Type 2 inflammation and airway remodelling - Fixed airflow limitation	- Response to anti-IL-13 (in asthmatic individuals with high levels of periostin)
**Procalcitonin**	Blood sample	Bacterial infection	- Shorter duration of antibiotic treatment, fewer antibiotics - Side effects and lower mortality when used as a tool to guide the prescription of antibiotics

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
