# Peer review of "Treatment Response Biomarkers in Asthma and COPD"

_diagnostics, 2021, doi:10.3390/diagnostics11091668_

Round 1

Reviewer 1 Report

I think this is a nice review but it would benefit from three major additions:

  1. Abbreviations should be defined. I will leave it up to authors to either define in text or have an appendix. Maybe have both. Appendix for something FEV1 but text for something like CALIMA and SIROCCO studies. Reader should not have to go elsewhere to look up these abbreviations.
  2. Drugs names should be the actual ingredient and possibly mechanism of action and the drug name maybe in parenthesis. For example mepolizumab is not as critical to include as the actual ingredient and it's mechanism of action. There are so many new drugs out there making their brand name not very informative in a scientific review.
  3. The review would benefit from a flow chart. What is basically in text but a flow chart that can be referred to and includes makers, diagnosis, outcomes etc. Would make a great reference. It can even be color coded for treatment vs response vs marker or whatever the authors wish but a summary figure is lacking. 

I realize this will make reading this a little less straightforward but it will make it more informative and capable of standing alone. To make it easier some things that are not used again need not be referred to with their abbreviation. For example "randomized, clinical trials" does not need to be defined as (RCTs) if only used in that spot and no other.

Author Response

Dear Reviewer, 

We appreciate your comments and suggestions, which we have incorporated in the revised manuscript.

  1. We have now defined all abbreviations in the text, including acronyms for the large clinical trials. 

  2. We have only mentioned the active drug names ingredient and not the brand name, e.g. omalizumab, mepolizumab, dupilumab and not Xolair, Nucala, Dupixent, respectively.
    Further, we have now for each biological drug targeting the IL-5 pathway provided a parenthesis that very briefly explains the mechanism of action. 

  3. The review would benefit from a flow chart. What is basically in text but a flow chart that can be referred to and includes makers, diagnosis, outcomes etc. Would make a great reference. It can even be color coded for treatment vs response vs marker or whatever the authors wish but a summary figure is lacking.

    We have now added a table that provides an overview of the various biomarkers, method of measurements, what the biomarker indicates and predicts. 

Reviewer 2 Report

There are several remarks: 1. The authors have highlighted several subsections in the article, which at first glance look logical. Razlel 2 and 3 focus on markers of inflammation in asthma and COPD, respectively, but section 4 focuses on procalcitonin in both cases. I believe that section 4 should be divided and information should be added to the corresponding sections 2 and 3, as well as the word "inflammatory" should be removed from the titles of the sections, it would be more logical. 2. There is a very lack of a summary table of markers for asthma and COPD and their comparative analysis. 

Author Response

Dear Reviewer,

We are grateful for your suggestions and comments. 

1.
We indeed follow your idea of dividing Section 4 regarding Procalcitonin into Section 2 and 3. Although the manuscript focuses on asthma and COPD, Section 4 serves as a more general section including respiratory tract infections and pneumonia in patients without asthma or COPD. This purpose of this is to review the literature on the usefulness of procalcitonin as a tool to guide antibiotic treatment.  
We have, however, changed the title of the section to "Procalcitonin as a Tool to Guide Antibiotic Treatment for Respiratory Tract Infections" to indicate broader approach in this section, but with main focus on asthma and COPD. 

2. We have removed the word "inflammatory" from the title in the sections and added "biomarkers" instead. 

3. We have now added a summary table of biomarkers. 

Round 2

Reviewer 2 Report

I have no more comments on the article, in its present form it can be recommended for publication.